# VIOLA: Imitation Learning for Vision-Based Manipulation with Object Proposal Priors

**Yifeng Zhu**[1], **Abhishek Joshi**[1], **Peter Stone**[1,2], **Yuke Zhu**[1]

[1]The University of Texas at Austin [2]Sony AI

**Abstract:** We introduce VIOLA, an object-centric imitation learning approach to learning closed-loop visuomotor policies for robot manipulation. Our approach constructs object-centric representations based on general object proposals from a pre-trained vision model. VIOLA uses a transformer-based policy to reason over these representations and attend to the task-relevant visual factors for action prediction. Such object-based structural priors improve deep imitation learning algorithm's robustness against object variations and environmental perturbations. We quantitatively evaluate VIOLA in simulation and on real robots. VIOLA outperforms the state-of-the-art imitation learning methods by $45.8\%$ in success rate. It has also been deployed successfully on a physical robot to solve challenging long-horizon tasks, such as dining table arrangement and coffee making. More videos and model details can be found in supplementary material and the project website: https://ut-austin-rpl.github.io/VIOLA.

**Keywords:** Imitation Learning, Manipulation, Object-Centric Representations

## 1 Introduction

Vision-based manipulation is a critical ability for autonomous robots to interact with everyday environments. It requires the robots to understand the unstructured world through visual perception to determine intelligent behaviors. In recent years, deep imitation learning (IL) [1–4] has emerged as a powerful approach to training visuomotor policies on diverse offline data, particularly human demonstrations. Its success stems from the effectiveness of training over-parameterized neural networks end-to-end with supervised learning objectives. These models excel at mapping raw visual observations to motor actions without manual engineering. While deep IL methods often distinguish themselves from reinforcement learning counterparts in their scalability to long-horizon tasks, a burgeoning body of recent work pointed out that IL methods lack robustness to covariate shifts and environmental perturbations [5–11]. End-to-end visuomotor policies are likely to falsely associate actions with task-irrelevant visual factors, leading to poor generalization in new situations.

In this work, we endow imitation learning algorithms with awareness about objects and their interactions to improve their efficacy and robustness in vision-based manipulation tasks. As cognitive science studies suggest, explaining a visual scene as objects and their interactions facilitates humans to learn fast and make accurate predictions [12–14]. Inspired by these findings, we hypothesize that decomposing a visual scene into factorized representations of objects in the scenes would enable robots to reason about the manipulation workspace in a modular fashion and improve their generalization ability. To this end, we develop an *object-centric imitation learning* approach, which infuses structural object-based priors into the model architecture of visuomotor policies. Training policies with these priors would make it easier for the model to focus on task-relevant visual cues while discarding spurious dependencies.

The first and foremost challenge of such an object-centric approach is to determine what constitutes an *object* and how objects are represented. The definitions of objects are often fluid and task-dependent for manipulation tasks. This work studies the notions of objects operationally and considers them as disentangled visual concepts that inform the robot's decision-making. Previous works have explored learning visuomotor policies with awareness of objects, but they are limited to simple control domain [6], single object manipulation [15], or require costly annotations for object detection [16]. We are motivated by the recent advances in visual recognition, in particular, image models for generating object proposals [17, 18], *i.e.*, localized bounding boxes on 2D images. These object proposals capture general priors of "objectness" across appearance variations and object cate-

6th Conference on Robot Learning (CoRL 2022), Auckland, New Zealand.

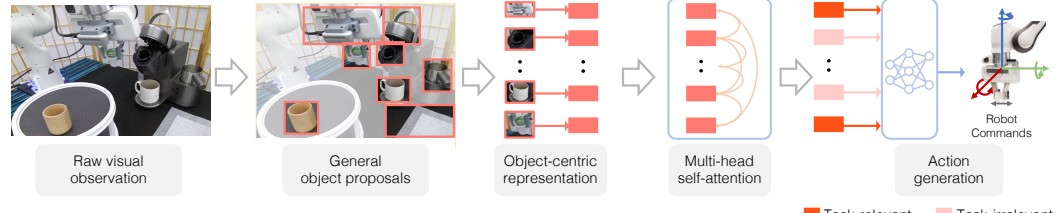

Figure 1: VIOLA first obtains a set of general object proposals from raw visual observations. It extracts object features from the proposals to build the object-centric representation. The transformer-based policy uses multi-head self-attention to reason over the representation and identify task-relevant regions for action generation.

gories. They have served as intermediate representations for downstream vision tasks, such as object detection and instance segmentation [19–21]. In this work, we investigate using object proposals from a pre-trained vision models as object-centric priors for visuomotor policy in manipulation and use the object proposals as a starting point to build our object-centric representations.

We introduce VIOLA (**V**isuomotor **I**mitation via **O**bject-centric **LeA**rning), an object-centric imitation learning model to train closed-loop visuomotor policies for robot manipulation. The high-level overview of the method is illustrated in Figure 1. VIOLA first uses a pre-trained Region Proposal Network (RPN) [18] to get a set of general object proposals from raw visual observations. We extract features from each proposal region to build the factorized object-centric representations of the visual scene. These object-centric representations are converted into a set of discrete tokens and processed by a Transformer encoder [22]. The transformer encoder learns to focus on task-relevant regions while ignoring the irrelevant visual factors for decision-making through the multi-head self-attention mechanism when trained on supervised imitation learning objectives.

We compare VIOLA against state-of-the-art deep imitation learning methods for vision-based manipulation in simulation and on a real robot. We use simulation to systematically evaluate the policies' performances and generalization abilities in the canonical setting (*i.e.*, testing in the training distribution) and under three challenging variations, including initial object placements, presence of distracting objects, and camera pose perturbations. Our quantitative evaluations show that VIOLA outperforms the most competitive baseline by $45.8\%$ in success rate. When visual variations such as jittered camera views are introduced, VIOLA maintains its robust behaviors of precise grasping and manipulation, while end-to-end learning methods would fail to reach the target objects. VIOLA also produces visuomotor policies to solve three challenging real-world tasks with a small set of 50 demonstrations, including a multi-stage coffee-making task where VIOLA achieves $60\%$ success rate while baseline methods fail entirely.

Our contributions with VIOLA are three-fold: 1) We learn object-centric representations based on general object proposals and design a transformer-based policy that determines task-relevant proposals to generate the robot's actions; 2) We show that VIOLA outperforms state-of-the-art baselines in simulation and validate the effectiveness of our model designs through ablative studies; and 3) We show that VIOLA learns policies on a real robot to complete challenging tasks.

## 2 Related Work

**Imitation Learning (IL) for Manipulation.** IL has been an established paradigm for acquiring manipulation policies for decades. It can be roughly categorized into non-parametric and parametric approaches. Non-parametric approaches, such as DMP and PrMP, can effectively acquire manipulation behaviors through a small number of demonstrations [23–26]. However, they typically focus on open-loop trajectory generation and fall short in handling high-dimensional observations. Parametric approaches, especially neural networks, have shown promise in vision-based manipulation. Nevertheless, these approaches are susceptible to distributional shifts and observation noises [1–4, 27]. Object-centric priors have been previously explored in imitation learning policies to overcome the issues above [15, 16]. However, these previous works either focus on the manipulation of single object instances, or requires costly annotations for pre-training object detections. Based on the same conceptual idea as previous object-centric imitation learning, VIOLA uses a pre-trained RPN to introduce object proposals as object-centric priors into the end-to-end IL policies, thus improving their robustness towards visual variations and solving tasks that involve complicated interaction with multiple objects.

**Visual Representations for Visuomotor Policies.** Various types of intermediate visual representations have been explored for visuomotor policy learning. Object bounding boxes have been commonly used as intermediate visual representations [28, 16, 29]. However, they require fine-tuned or category-specific detectors and cannot easily generalize to tasks with previously unknown objects. Recently, deep learning methods have enabled IL to train policies end-to-end on raw observations [2, 3]. These methods are prone to covariate shift and causal confusion [6], resulting in poor generalization performances. Similar to our work, a large body of literature has looked into incorporating additional inductive biases into end-to-end policies. Notable ones include spatial attention [30–32] and affordances [33–38]. However, these representations are purposefully designed for specific motion primitives, such as pick-and-place, limiting their abilities to generate diverse manipulation behaviors.

**Object-Centric Representation.** Object-centric representations have been widely used in visual understanding and robotics tasks, where researchers seek to reason about visual scenes in a modular way based on the objects presented. In robotics, poses [39–41] or bounding boxes [28, 16, 29] are commonly used as object-level abstractions. These representations often require prior knowledge about object instance/category and do not capture fine-grained details, falling short in applying to new tasks without previous unknown objects. Unsupervised object discovery methods [42, 43] learn object representation without manual supervision. However, they fall short in handling complex scenes [42, 43], hindering the applicability to realistic manipulation tasks. Recent work from the vision community has made significant progress in generating object proposals for various downstream tasks, such as object detection [19–21] and visual-language reasoning [44, 45]. Motivated by the effectiveness of region proposal networks (RPNs) on out-of-distribution images [21], we use object proposals to scaffold our object-centric representations for robot manipulation tasks.

## 3   Approach

We introduce VIOLA, an object-centric imitation learning approach to vision-based manipulation. The core idea is to decompose raw visual observations into object-centric representations, on top of which the policy generates closed-loop control actions. Figure 2 illustrates the pipeline. We first formulate the problem of visuomotor policy learning and describe two key components of VIOLA: 1) how we build the object-centric representations based on general object proposals, and 2) how we use a transformer-based architecture to learn policy over the object-centric representations.

### 3.1   Problem Formulation

We formulate a robot manipulation task as a discrete-time Markov Decision Process, which is defined as a 5-tuple: $\mathcal{M} = (\mathcal{S}, \mathcal{A}, \mathcal{P}, R, \gamma)$, where $\mathcal{S}$ is the state space, $\mathcal{A}$ is the action space, $\mathcal{P}(\cdot|s, a)$ is the stochastic transition probability, $R(s, a, s')$ is the reward function, and $\gamma \in [0, 1)$ is the discount factor. In our context, $\mathcal{S}$ is the space of the robot's raw sensory data, including RGB images and proprioception, $\mathcal{A}$ is the space of the robot's motor commands, and $\pi : \mathcal{S} \rightarrow \mathcal{A}$ is a closed-loop sensorimotor policy that we deploy on the robot to perform a task. The goal of learning a visuomotor policy for robot manipulation is to learn a policy $\pi$ that maximizes the expected return $\mathbb{E}[\sum_{t=0}^{\infty} \gamma^t R(s_t, a_t, s_{t+1})]$.

In our work, we use behavioral cloning as our imitation learning algorithm. We assume access to a set of N demonstrations $D = \{\tau_i\}_{i=1}^{N}$, where each trajectory $\tau_i$ is demonstrated through teleoperation. The goal of our behavioral cloning approach is to learn a policy that clones the actions from demonstrations $D$.

We aim to design an object-centric representation that factorizes the visual observations of an unstructured scene into features of individual entities. For vision-based manipulation, we assume no access to the ground-truth states of objects. Instead, we use the top $K$ object proposals from a pre-trained Region Proposal Network (RPN) [18] to represent the set of object-related regions of an image. These proposals are grounded on image regions and optimized for covering the bounding boxes of potential objects. We treat these $K$ proposals as the $K$ (approximate) objects and extract visual and positional features to build *region features* from each proposal. For manipulation tasks, reasoning about object interactions is also essential to decide actions, To provide contextual information for such relational reasoning, we design three *context features*: a global context feature from the workspace image to encode the current stage of the task; an eye-in-hand visual feature from the eye-in-hand camera to mitigate occlusion and partial observability of objects; and a proprioceptive feature from the robot's states. We call the set of region features and context features at time step $t$ as the

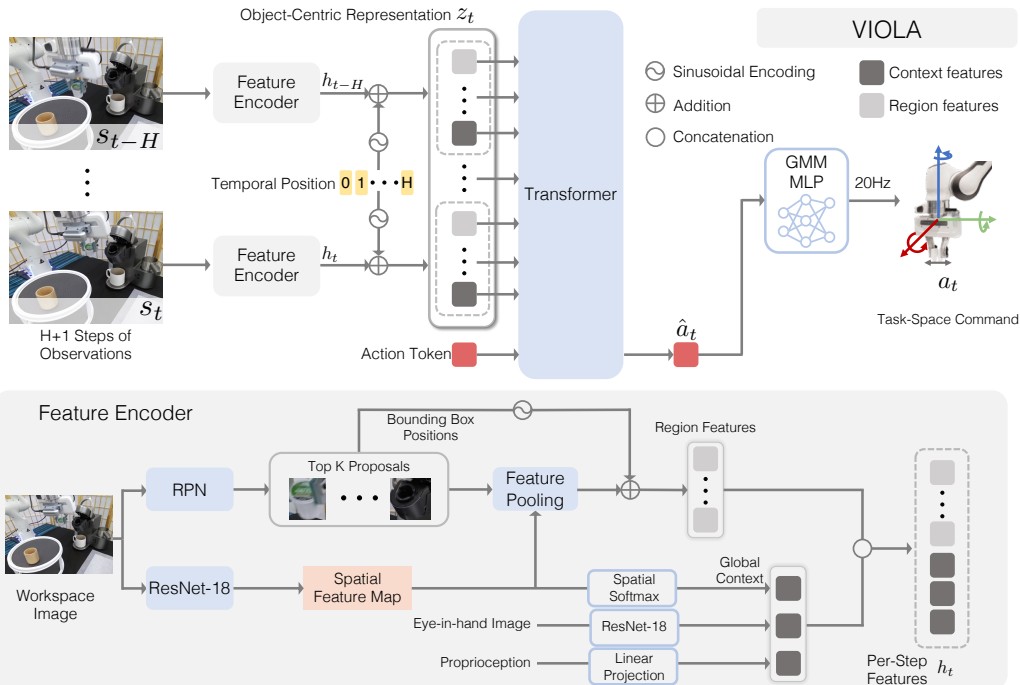

Figure 2: Model architecture of VIOLA. At time $t$, it computes the per-step features $h_t$ using the top $K$ object proposals. Then it constructs the object-centric representation $z_t$ by composing per-step features from the last $H+1$ time steps with their temporal positional encodings. Transformer-based policy reasons over $z_t$ to output a latent vector of the action token, $\hat{a}_t$, which is passed through a multi-layer perceptron (MLP) to generate actions.

*per-step feature* $h_t$. Finally, our object-centric representation $z_t$ is a composition of per-step features from the last $H+1$ time steps with their temporal positional encoding, as we describe in Sec. 3.2.

## 3.2 Object-Centric Representation

This section describes how to build the object-centric representation $z_t$. VIOLA first obtains general object proposals using a pre-trained RPN. VIOLA computes region features from proposals and obtains a per-step feature $h_t$ using the region features and the three context features which we describe next. VIOLA builds $z_t$ through the temporal composition of $h_t$ from a history of observations.

**General Object Proposals.** At each time step $t$, we generate object proposals using a pre-trained RPN on the workspace image. We select the top $K$ proposals with the highest confidence scores, which indicate regions that contain objects with the highest likelihood. The intuition of using a pre-trained RPN is that it captures good priors of "objectness" in RGB images through the supervision of natural image datasets. Recent work shows that an RPN trained on an 80-class dataset can generalize to a 2000-class dataset [21]. Our preliminary studies suggested that such a generalization ability also holds for localizing objects on raw images in our simulation and real-world tasks regardless of the domain gaps. We use the pre-trained RPN from Zhou et al. [21] to localize regions with objects.

**Region Features.** For our policy to reason over objects and their spatial relations, we need to identify what objects each region contains and where these regions are from the top $K$ proposals. To encode this information, we design a *visual feature* and a *positional feature* for each region. To extract the visual feature from a region, we learn a spatial feature map by encoding the workspace image with the ResNet-18 module [46] and extract the visual feature using ROI Align [17]. We use a learned spatial feature map for visual features rather than from a pre-trained feature pyramid network in detection models because we share the same objective of localizing objects as object detectors but different goals for the downstream tasks — pre-trained feature pyramid networks are optimized for visual recognition tasks, but we need actionable visual features that are informative for continuous control. Such a design choice is supported by our ablation experiment in Appendix B.2 For positional features, we encode the coordinates of bounding box corners on images using sinu-

soidal positional encoding [22] (more details in Appendix A). We flatten each visual feature and add it to the positional feature of the same region to obtain a region feature.

**Context Features.** We extract the region features for the policies to reason over individual objects. However, they are insufficient for decision-making in vision-based manipulation tasks, so we introduce three context features to assist decision-making. As regions only encode local information, we add a *global context feature* to encode the current task stage from observation, which is computed from the spatial feature map of the workspace image using Spatial Softmax. During manipulation, the robot's gripper often occludes objects in the workspace view, so we add an *eye-in-hand feature* from eye-in-hand images to represent the occluded objects. We also encode the robot's measurements of its joints and the gripper into *proprioceptive features* for the policy to generate precise actions based on the robot's states. We aggregate the context and region features at each time step $t$ into a set, which we refer to as per-step feature $h_t$.

**Temporal Composition.** We build object-centric representation $z_t$ through the temporal composition of $h_t$ that captures temporal dependencies and dynamic changes of object states. Building policies over a sequence of past observations rather than the most recent observation has been shown effective by prior work [1, 27]. In our work, the temporal composition also increases the recall rate of object proposals, making the policy more robust in the case of detection failures. More specifically, $z_t$ is built from a set of per-step features $\{h_{t-i}\}_{i=0}^{H}$ from the last $H + 1$ time steps. To encode their temporal ordering, we add sinusoidal position encoding of the temporal positions $\{PE_i\}_{i=0}^{H}$ to the per-step features, resulting in our object-centric representation $z_t = \{h_{t-i} \oplus PE_i\}_{i=0}^{H}$. Our ablative studies (see Sec. 4) indicate the importance of temporal positional encoding that retains the temporal ordering of features, especially when using our transformer-based policy.

### 3.3 Transformer-based Policy

We desire a policy that focuses on task-relevant region features in $z_t$ to generate actions. Regions that associate with task-relevant objects facilitate the accurate prediction of actions, while regions with task-irrelevant objects are likely to confound the policies. We seek to use a transformer [22] as the policy backbone, which allows policies to reason over objects and their relations with its self-attention mechanism. The core of a transformer is an encoder layer which consists of a multi-head self-attention block (MHSA), a layer-normalization [47] function, and a fully-connected neural network (FFN). A transformer encoder layer takes as input a sequence of $n$ latent vectors $(x_1, \ldots, x_n)$ (also called *tokens*) and outputs a sequence of $n$ latent vectors $(y_1, \ldots, y_n)$. A MHSA block consists of multiple self-attention blocks in parallel, which computes attention over all the tokens and computes a weighted sum of token values. We refer readers to Vaswani et al. for more details [22]. In short, each transformer encoder outputs $Y = \text{FFN}(\text{LayerNorm}(\text{MSHA}(X))$, where each row of $Y$ is an output latent vectors $y_i$ that corresponds to $x_i$. Our transformer-based policy is a stack of multiple transformer encoder layers, which allows for a higher degree of expressiveness over input tokens compared to a single layer.

For our policy, we tokenize our object-centric representation $z_t$, treating each region and context feature vector as an input token. To make action generation attend more to task-relevant region features than task-irrelevant ones, we append a learnable token, action token, to the input sequence of a transformer. The action token design resembles the specific class tokens in natural language understanding tasks [48] or visual recognition tasks [49], which are used for outputting latent vectors for downstream tasks. Similarly, we can get the output latent vector $\hat{a}_t$ from the action token, which learns to attend to task-relevant regions through action supervision. In the end, we pass $\hat{a}_t$ through a two-layered fully-connected network, followed by a GMM (Gaussian Mixture Model) output head, which has been shown effective to capture the diverse multimodal behaviors in demonstration data [50, 27].

**Implementation Details.** Here we describe several key implementation details. Full implementation details are in Appendix A. We set $K = 20$ in simulation, which offers a nice trade-off between object recall rates and computation efficiency (see Appendix B.1). In the real world, we set $K = 15$ to attain a high recall rate on real images. For temporal composition, we use $H = 9$ to be consistent with the setup in BC-RNN [27]. The action token is a learnable parameter randomly initialized from the normal distribution.

We use color augmentation and pixel shifting [4, 27, 49] during training to encourage the generalization ability of policies. Additionally, we adopt random erasing [51] to prevent policies from overfitting to specific region features. Concretely, we randomly apply random erasing during

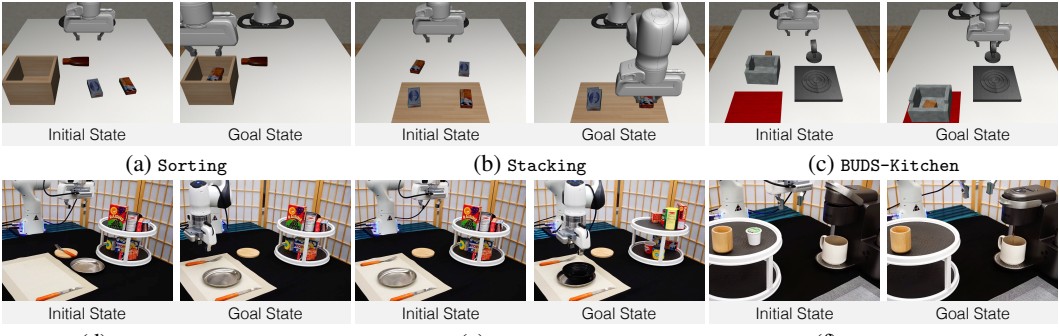

Figure 3: Visualization of initial and goal configurations for the simulation and real-world tasks.

| Tasks | Variants | BC [54] | OREO [6] | BC-RNN [27] | VIOLA-Patch | VIOLA |
|---|---|---|---|---|---|---|
| Sorting | Canonical | $25.1 \pm 1.6$ | $38.7 \pm 0.5$ | $62.8 \pm 0.9$ | $71.2 \pm 1.0$ | $\mathbf{87.6} \pm 1.1$ |
|  | Placement | $1.9 \pm 0.7$ | $8.3 \pm 1.7$ | $11.7 \pm 1.0$ | $48.5 \pm 2.2$ | $\mathbf{68.3} \pm 1.5$ |
|  | Distractor | $14.5 \pm 1.9$ | $26.0 \pm 11.4$ | $46.7 \pm 6.5$ | $58.6 \pm 4.2$ | $\mathbf{74.4} \pm 5.7$ |
|  | Camera-Jitter | $6.1 \pm 0.3$ | $16.3 \pm 3.3$ | $9.6 \pm 0.4$ | $34.6 \pm 1.4$ | $\mathbf{50.7} \pm 0.6$ |
| Stacking | Canonical | $14.9 \pm 1.1$ | $13.3 \pm 2.0$ | $27.8 \pm 0.8$ | $71.2 \pm 1.0$ | $\mathbf{71.3} \pm 1.0$ |
|  | Placement | $1.6 \pm 0.3$ | $0.3 \pm 0.5$ | $0.0 \pm 0.0$ | $28.4 \pm 1.9$ | $\mathbf{46.7} \pm 0.2$ |
|  | Distractor | $5.6 \pm 1.4$ | $5.6 \pm 4.5$ | $14.4 \pm 3.2$ | $\mathbf{41.4} \pm 5.1$ | $38.6 \pm 2.8$ |
|  | Camera-Jitter | $2.1 \pm 0.3$ | $0.6 \pm 0.9$ | $1.0 \pm 0.0$ | $17.6 \pm 1.0$ | $\mathbf{29.3} \pm 1.7$ |
| BUDS-Kitchen | Canonical | $0.0 \pm 0.0$ | $0.0 \pm 0.0$ | $0.0 \pm 0.0$ | $5.8 \pm 0.6$ | $\mathbf{84.2} \pm 1.3$ |
|  | Placement | $0.0 \pm 0.0$ | $0.0 \pm 0.0$ | $0.0 \pm 0.0$ | $3.1 \pm 0.6$ | $\mathbf{58.4} \pm 1.1$ |
|  | Distractor | $0.0 \pm 0.0$ | $0.0 \pm 0.0$ | $0.0 \pm 0.0$ | $10.7 \pm 0.7$ | $\mathbf{73.2} \pm 6.2$ |
|  | Camera-Jitter | $0.0 \pm 0.0$ | $0.0 \pm 0.0$ | $0.0 \pm 0.0$ | $2.6 \pm 0.3$ | $\mathbf{41.2} \pm 1.7$ |

Table 1: Success rates (%) in simulation setups over 100 initializations with repeated runs of 3 random seeds.

training with a probability of $0.5$. Random erasing fills in Gaussian noise into a randomly selected region whose size is $2\%$ to $5\%$ of the image with an aspect ratio ranging from $0.5$ to $1.5$.

## 4 Experiments

We design our experiments to answer the following questions: 1) How well does VIOLA perform against state-of-the-art end-to-end imitation learning algorithms? 2) How does it take advantage of object-centric representations? 3) What design choices are essential for good performance? and 4) Is VIOLA practical for real-world deployment?

### 4.1 Experimental Setup

We conduct quantitative evaluations in simulation and real-world tasks to validate our approach. The simulation tasks are designed with the robosuite [52] framework and used for quantitative comparisons between VIOLA and baselines. We also validate our design choices through ablative studies. We design three simulation tasks, Sorting, Stacking, BUDS-Kitchen, and three real-world tasks, Dining-PlateFork, Dining-Bowl, Make-Coffee that cover a rich spectrum of manipulation behaviors that combine prehensile and non-prehensile motions. We visualize their initial and goal configurations in Figure 3. We design these tasks to understand if the use of object priors benefits policy learning along two axes of task characteristics: large placement variations of objects and multi-stage long-horizon execution. For the first axis, we design Sorting and Stacking to have large initial ranges of object placements. For the second axis, we use BUDS-Kitchen, a multi-stage long-horizon manipulation task from prior work [4]. The real-world tasks are designed to resemble real-world everyday tasks: Dining-PlateFork and Dining-Bowl for dining table arrangement, Make-Coffee for espresso coffee making. Full details of tasks and data collection are included in Sec. C.

**Evaluation Setup.** To systematically evaluate the efficacy and robustness of the learned policies, we design the following testing setups in simulation: 1) Canonical: all the objects and sensor configurations follow the same distribution as seen in demonstrations; and 2) Three testing variants, namely Placement, Distractor, and Camera-Jitter. The design of these variants is inspired by the MAGICAL benchmark [53] for evaluating systematic generalization of imitation learning policies. The design principle of testing variants is to keep the task semantics the same as in the canonical setup. At the same time, we identify the three challenging axes of variations for learning-based manipulation, namely initial object placements (Placement), distractor objects presented in the scene (Distractor), and camera pose jitters (Camera-Jitter).

| Models | Canonical | Placement | Distractor | Camera-Jitter |
|---|---|---|---|---|
| Base | $0.0 \pm 0.0$ | $0.0 \pm 0.0$ | $0.0 \pm 0.0$ | $0.0 \pm 0.0$ |
| + Temporal Observation | $\uparrow 69.5 \pm 2.5$ | $\uparrow 25.1 \pm \uparrow 2.2$ | $\uparrow 37.7 \pm 3.4$ | $\uparrow 39.9 \pm 1.5$ |
| + Temporal Positional Encoding | $\uparrow 74.0 \pm 0.8$ | $\uparrow 48.3 \pm 1.9$ | $\uparrow 48.3 \pm 3.3$ | $\uparrow 50.9 \pm 1.5$ |
| + Region Visual Features | $72.8 \pm 0.9$ | $\downarrow 37.7 \pm 1.2$ | $\uparrow 54.6 \pm 4.6$ | $49.2 \pm 1.6$ |
| + Region Positional Features | $\uparrow 80.2 \pm 2.9$ | $38.6 \pm 0.3$ | $\uparrow 62.0 \pm 5.7$ | $\downarrow 46.5 \pm 1.9$ |
| + Random Erasing (=VIOLA) | $\uparrow 87.6 \pm 1.1$ | $\uparrow 68.3 \pm 1.5$ | $\uparrow 74.4 \pm 5.7$ | $\uparrow 50.7 \pm 0.6$ |

Table 2: The effect of VIOLA model designs on success rates (%) in `Sorting` task. Changes larger than 2% are annotated with $\uparrow$ / $\downarrow$.

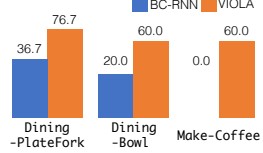

Figure 4: Success rates (%) in real robot tasks.

## 4.2 Experiment Results.

We answer question (1) by quantitatively comparing VIOLA with the following baselines: **BC**: behavioral cloning that conditions on current observations; **OREO** [6]: behavioral cloning method that learns object-aware discrete codes using VQ-VAE [55] and learns policies thereon; **BC-RNN** [27, 56, 15, 57]: a state-of-the-art method that uses a temporal sequence of past observations with recurrent neural networks. **VIOLA-Patch**: a variant of VIOLA, where we use a regular grid of image patches as inputs to the policy rather than proposals, similar to ViT [58]. To make a fair evaluation, we randomly generate 100 initial configurations and repeat runs with three different random seeds. We evaluate all policies on the same set of pre-generated initial configurations.

Table 1 shows that VIOLA outperforms the most competitive baseline BC-RNN by $50.8\%$ success rate in `Canonical` and $44.1\%$ in three testing variants. VIOLA's strong performance suggests the advantage of object-centric representations for visuomotor imitation. OREO's results suggest that learning object-aware discrete codes via unsupervised learning does not consistently improve performance over simple BC for all tasks. The comparisons between BC, OREO, and the other methods using temporal windows show the importance of temporal modeling for these algorithms to achieve high performances and robustness in complex vision-based manipulation domains.

Table 1 shows that VIOLA-Patch has comparable performance to VIOLA in some evaluation setups. It shows that the transformer backbone can attend to patch regions that cover task-relevant visual cues, even though the patch division is agnostic to objects. Nonetheless, VIOLA still performs better than VIOLA-Patch, especially in the long-horizon task `BUDS-Kitchen`, suggesting regions with complete coverage of objects is critical for the success of VIOLA, highlighting the importance of the object priors.

**Ablative Studies.** To answer questions (2) and (3), we use ablative studies to validate our model's design and show how it takes advantage of the object-centric priors. Table 2 quantifies the effects of the transformer backbone, the use of temporal windows, object proposal regions, and our data augmentation technique. We start with our base model, a transformer model that only takes the current observations as input. Second, we add the temporal window of observations similar to BC-RNN. Results show that the use of temporal observation is key to unleashing the power of the transformer architecture, making the performance of this ablated version comparable to BC-RNN. Nonetheless, this model does not encode temporal ordering as the transformer model is invariant to input permutation. In the next row, we add temporal positional encoding [22] to the input sequence, which produces a method that outperforms the top baselines.

In the next two rows, we procedurally add visual and positional features of regions to prove that our model does exploit the object priors. Results show that visual features alone only improve our model's robustness to `Distractor`. When using both visual and positional features, the model performs better in `Canonical` and `Distractor`, but worse in `Placement` and `Camera-Jitter`. We hypothesize that this ablated version overfits to locations of proposal boxes in demonstrations; therefore, it generalizes worse in `Placement` and `Camera-Jitter` where the distribution of object locations on 2D images shifts from demonstrations. To mitigate such overfitting, we use Random Erasing [51] to achieve the highest success rates in all evaluation setups.

**Real Robot Evaluation.** We compare VIOLA against the SOTA baseline, BC-RNN, on all three real-world tasks. We evaluate in 10 different initial configurations and repeat 3 times in the same configuration for each policy evaluation. To mitigate potential human bias introduced by setting up objects before each trial, we employ the A/B testing paradigm for evaluation. That is, we reset the environment while being agnostic to the next policy to execute. The entire evaluation process for one initialization is repeated until all evaluation trials are completed.

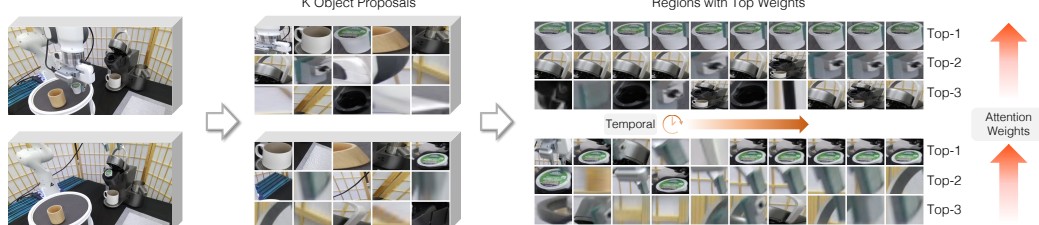

Figure 5: Visualization of transformer attention over regions. When the robot is to grasp the k-cup, VIOLA's top attention is over the k-cup across observations while it takes the robot fingers and coffee machine into account to help spatial reasoning. When the robot is to close the coffee machine, VIOLA's top-1 attention is all over k-cups, the robot gripper, and the coffee machine, and it reasons over these regions to generate actions.

The quantitative evaluation in Figure 4 shows that VIOLA outperforms BC-RNN by $46.7\%$ success rate on average. Qualitatively, we observe that VIOLA can robustly grasp k-cups or open the coffee machine in the Make-Coffee task, while BC-RNN tends to reach to wrong positions where grasps are missed or to release the gripper mistakenly. In the supplementary videos, we include the evaluation rollouts of VIOLA for all the tasks. We also qualitatively visualize the attention of VIOLA in Figure 5 by showing the top 3 regions with the highest attention weights at each timestep of temporal observation. The figure shows that when the robot is to grasp the k-cup, VIOLA's top attention is over the k-cup while it also takes the robot fingers and coffee machine into account to facilitate spatial reasoning. When the robot is to close the coffee machine, VIOLA's top-1 attention spreads over the k-cup, robot gripper, and the coffee machine, and the policy generates actions for closing through spatial reasoning over these regions.

## 5 Conclusion

We present VIOLA, an object-centric imitation learning approach to learning closed-loop visuo-motor policies for robot manipulation. Our approach uses general object proposals to build the object-centric representation. This representation encodes the visual and positional features from the proposal regions and context features of global scene information and robot states. We design a transformer-based policy to identify task-specific relevant regions for action generation. The results show the superior performances of VIOLA compared to state-of-the-art baselines in simulation and the real world. We also validate our model designs through ablative studies, showing how each model component impacts policy performance.

**Limitations.** While VIOLA has achieved great success in learning robust visuomotor policies, it suffers from common limitations of learning from offline demonstration datasets. One future direction is to use this object-centric representation in an online manner so that it can improve over roll-out experience over time. Also, we use a pre-trained RPN without adaptation, which may fail to generalize to manipulation scenes with aggressive distributional shifts from natural images on which the RPN is trained. A possible remedy is to fine-tune the RPN while training the policy. Another limitation of our approach is that we only consider monocular RGB images in the input data, which lacks the 3D information of objects. This formulation prevents VIOLA from excluding the background visual elements in the input, which would limit the generalization ability of VIOLA in variants of large visual changes, such as table texture change. For future work, we would like to extend VIOLA to using depth images so that distracting background elements such as tables can be disentangled from the object representations.

**Acknowledgments**
The authors would like to thank Yue Zhao for the insightful discussion on the project and the manuscript. This work has taken place in the Robot Perception and Learning Group (RPL) and Learning Agents Research Group (LARG) at UT Austin. RPL research has been partially supported by the National Science Foundation (CNS-1955523, FRR-2145283), the Office of Naval Research (N00014-22-1-2204), and the Amazon Research Awards. LARG research is supported in part by NSF (CPS-1739964, IIS-1724157, FAIN-2019844), ONR (N00014-18-2243), ARO (W911NF-19-2-0333), DARPA, GM, Bosch, and UT Austin's Good Systems grand challenge. Peter Stone serves as the Executive Director of Sony AI America and receives financial compensation for this work. The terms of this arrangement have been reviewed and approved by the University of Texas at Austin in accordance with its policy on objectivity in research.

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
