# OpenReview forum: "VIOLA: Object-Centric Imitation Learning for Vision-Based Robot Manipulation"
_robot-learning.org/CoRL/2022/Conference — CoRL 2022 Poster_

### Official Review · Reviewer_n5Tr · 2022-07-06

**Originality:** Very Good
**Technical Quality:** Excellent
**Clarity Of Presentation:** Excellent
**Impact:** 3

**Recommendation:**

Strong Accept: I recommend accepting the paper and will argue for my recommendation even if other reviewers hold a different opinion.

**Summary:**

The paper presents a transformer-based object-centric Visual Imitation Learning method, topping relevant state-of-the-art methods. The method is shown in several simulated tasks, introduced in the paper, including a multi-stage environment, and in the real world. Ablations show that the method is able to effectively deal with perturbations, jitter and object placement variations.

**Issues:**

Technical issues and questions:
- Hand-eye camera images are used as input to deal with local occlusion, but this choice is not discussed in detail. Would it be self-sufficient to have only the eye-in-hand camera? How much does the agent benefit from it being there?
- Random erasing, which seems to be key to using regional features, could use more explanation within the background/method of the paper.
- As per L151-L154, the model would be relying on a trainable RPN instead of a frozen one. Is the ResNet also not pre-trained? Wouldn’t this reduce the ability of the agent to generalize across similar objects (e.g., trained on one screwdriver, and then manipulating another). Moreover, it would be interesting to compare the frozen networks versus the trainable one used in the paper’s ablations (rather than in supplementary materials).
- Choice of ResNet18 over other backbones itself is unclear – modern transformer models might provide better features, it would have been interesting to see.

Minor issues:
- L20: Typo: “its success of it”
- Related Works, the cited Sieb et al. [24] has relevance in more than just the bounding box representation it uses since its very closely related to the proposed method, in that it is also vision-based object-centric imitation learning and the given summary of it doesn’t do it justice.
- L187: MHSA is redefined


**Quality Of The Limitations Section:**

Limitations are addressed clearly

**Reviewer Expertise:**

4: The reviewer is confident but not absolutely certain that the evaluation is correct

**Robotics Focus:**

Sufficient demonstration on hardware

**Strengths And Weaknesses:**

Strengths:
- This work is very good and shows a significant improvement over previous works.
- The design of the method is simple and elegant, with ablations justifying most design choices
- Clear and well presented

Weaknesses:
- The works lacks experimentation with more practical concerns such as generalization across similar objects, severe occlusion, and even more significant distractors than the ones shown.
- A concern is that the distractors shown in the supplementary files are minimal.


**Summary Of Recommendation:**

I have recommended strong accept, because I am willing to argue for this paper’s acceptance, and believe it is a positive contribution dealing with an important problem.

---

Update: Most of the concerns raised were answered by the authors. My evaluation remains unchanged as a Strong Acceptance.

---

> ### Author Response · Authors · 2022-08-24
> **Response (1/2)**
>
> Thank you very much for your thoughtful review and positive comments! We would like to address the comments from the reviewer. All the line numbers refer to the revised pdf to avoid line number mismatch from the original paper, and all the revised parts in the pdf are highlighted in red.
>
> > “The works lacks experimentation with more practical concerns such as generalization across similar objects, severe occlusion, and even more significant distractors than the ones shown.” and  “A concern is that the distractors shown in the supplementary files are minimal.”
>
> We agree with the reviewer that such generalizations are crucial for deployable vision-based robotics manipulation. Here we discuss each aspect of generalization below:
>
> 1. Similar objects: VIOLA can generalize to objects with close visual appearances, as we’ve demonstrated in our **Make-Coffee** tasks where different colors of K-cups are used during experiments. If the goal is to generalize across instances with large visual texture or geometry differences, it would require new model designs.
>
> 2. Distractors: We have mentioned in the limitation section (see Ln 331 - Ln 333) that our model would not handle drastic visual changes, including significant distractors. However, we would like to point out that VIOLA outperforms the state-of-the-art models in the presence of distractors (see Table 1). The distractors shown in the experiments, though small, are already very challenging to existing end-to-end policy models, as we've seen the results for all BC baselines, but VIOLA has outperforming taks success rates in it. A future direction for generalizing to significant distractors could be recognizing object categories to prune out all irrelevant objects.
>
> 3. Severe occlusion: As our approach focuses on object-centric learning for vision-based manipulation, it requires that the objects are visible in visual observations. If there are severe occlusions, the features of task-relevant objects would not be extracted from the observation. One potential extension is to use depth data to reconstruct the object geometry from partial observations of the objects. Not having used depth information is one of the limitations we have discussed in the limitation section, and we would like to consider this axis of generalization in future work.
>
> Extending generalization along all these axes will require model innovations, which could become follow-up papers in the future.
>
> > “Hand-eye camera images are used as input to deal with local occlusion, but this choice is not discussed in detail. Would it be self-sufficient to have only the eye-in-hand camera? How much does the agent benefit from it being there?”
>
> Thanks for the great question! We use the hand-eye camera images following the same setup as in the previous work on imitation learning for vision-based manipulation [1]. The last paragraph of Section 4.3 in [1] states that policies often benefit significantly from eye-in-hand images. Such evidence is also independently supported by other works [2, 3].
>
> Nonetheless, it is insufficient to have only eye-in-hand cameras. For example, one stage of our **Sorting** task requires the robot to grasp one box while moving to the exact location of the other box. Sometimes the robot’s eye-in-hand camera only captures the visual texture of in-hand objects after grasping at certain locations of the object. In this case, the robot does not have information about the other object's location from observations if the robot does not have the workspace images. We’ve added the rationale behind using eye-in-hand cameras in the text (see Ln 510 - Ln 511).
>
> In conclusion, an eye-in-hand camera is crucial to good performance, yet it is insufficient.

---

> > ### Author Response · Authors · 2022-08-24
> > **Response (2/2)**
> >
> > > “Random erasing, which seems to be key to using regional features, could use more explanation within the background/method of the paper.”
> >
> > We thank the reviewer for this suggestion. We have revised the text to include more explanation of random erasing in the paper (see Ln 226-228).
> >
> >
> > > “As per L151-L154, the model would be relying on a trainable RPN instead of a frozen one. Is the ResNet also not pre-trained? Wouldn’t this reduce the ability of the agent to generalize across similar objects (e.g., trained on one screwdriver, and then manipulating another). ”
> >
> > We would like to clarify that the RPN is a frozen one instead of a trainable one, and we revised the text for better clarity (see Ln 165 - Ln 166). In principle, the RPN can be fine-tuned, as we mentioned in our limitation sections (see Ln 329). In practice, we didn’t fine-tune because we found the recall rates of object proposals are high enough, and fine-tuning RPN proposals to crop every new object instances would introduce extra human annotations on images.
> >
> > As for ResNet, it is not pre-trained. The pre-trained ResNet does not bring benefits of generalization across similar objects as long as the new instance is different in visual appearance from the objects seen during the demonstration. Generalization across similar objects requires a successful behavioral cloning policy to take visual embeddings of two similar objects and generate the same action distributions. However, if the demonstration data does not cover similar object instances, the policy would not be optimized for the state action distributions of similar object instances, thus introducing distribution mismatch for imitation learning policies. Such a distribution mismatch would result in poor policy performance.
> >
> > > “ Moreover, it would be interesting to compare the frozen networks versus the trainable one used in the paper’s ablations (rather than in supplementary materials).”
> >
> > We thank the reviewer for appreciating the experiment of frozen networks versus the trainable ones. While this experiment is important to support that using trainable image encoders leads to better policy performance, we think it is orthogonal to the four research questions we brought up (Ln 230 - Ln 233), so we do not include the experiment details in the main text, but we added a pointer to this ablation study in Ln 169 of the main text.
> > > “Choice of ResNet18 over other backbones itself is unclear – modern transformer models might provide better features, it would have been interesting to see.”
> >
> > We chose ResNet-18 following the previous works on imitation learning for vision-based manipulation [1, 4]. We think the reviewer brings up a very insightful question about the potential of transformer-based backbones, so we ran an additional experiment using the ViT transformer backbone based on a widely-used [implementation](https://github.com/rwightman/pytorch-image-models/blob/master/timm/models/vision_transformer.py) in replacement of ResNet-18. The transformer-based backbone yields 80 $\pm$ 0.0 %, which is about 7% lower than our reported result with ResNet-18. This comparison result shows that ResNet-18 remains a good design for visual backbones as it provides better features than the modern transformer models.
> >
> >
> > > “Related Works, the cited Sieb et al. [24] has relevance in more than just the bounding box representation it uses since its very closely related to the proposed method, in that it is also vision-based object-centric imitation learning and the given summary of it doesn’t do it justice.”
> >
> > We thank the reviewer for pointing this out. We have added additional descriptions in Ln 83 - Ln 85 to reflect its relevance in vision-based, object-centric imitation learning.
> >
> >
> > > “L20: Typo: “its success of it” and “ L187: MHSA is redefined”
> >
> > Thank you very much! We’ve revised the text accordingly (see Ln 20, Ln 201).
> >
> > References:
> >
> > [1]What Matters in Learning from Offline Human Demonstrations for Robot Manipulation. Mandlekar et al. 2021
> >
> > [2]Vision-Based Manipulators Need to Also See From Their Hands. Hsu et al. 2022
> >
> > [3] Eyes on the Prize: Improved Percpetion for Robust Dynamic Grasping. Burgess-Limerick et al. 2022
> >
> > [4] GTI: Learning to Generalize Across Long-Horizon Tasks from Human Demonstrations. Xu et al. 2020

---

### Official Review · Reviewer_77ch · 2022-07-16

**Originality:** Good
**Technical Quality:** Good
**Clarity Of Presentation:** Very Good
**Impact:** 3

**Recommendation:**

Weak Reject: I recommend rejecting the paper, but will not argue for my recommendation if the majority of other reviewers have a different opinion.

**Summary:**

The paper presents an object-centric imitation learning approach to learning closed-loop visuomotor policies. The policy relies on object region proposal and features and a transformer based policy, as network structure, to improve generalization across view points, initializations and distractors. The policy outperform naive baselines and several ablated methods in both simulation and the real world.

**Issues:**

The architecture design seems to contribute a lot to the success rates in the network ablation study. I wonder if the performance gap to BC-RNN is still big if we vary the number of demonstrations or increase the BC network size.

Similarly, I wonder if the context information from spatial feature map, eye-in-hand image, and proprioception also provided to the simple BC baselines. I.e. how much the region proposals help with the transformer policy, v.s. How much the transformer policy outperforms vanilla networks.

The fact that random erasing helps a lot in table 2 is unclear to me. Is it because that the policy is overfitting to the K patches and some random masking would help with generalization?


**Quality Of The Limitations Section:**

Limitations are addressed clearly

**Reviewer Expertise:**

4: The reviewer is confident but not absolutely certain that the evaluation is correct

**Robotics Focus:**

Sufficient demonstration on hardware

**Strengths And Weaknesses:**

Strength:

The paper introduces an interesting object-centric approach for closed-loop visuomotor policy learning, and demonstrates robustness across viewpoints, initializations, and distractors under systematic simulation and real world evaluations.

Compared to the naive end-to-end approaches, VIOLA is more structured and makes use of transformers to attend to task relevant parts in the spatial-temporal domains.

The implementation details for making the transformer policy to work with temporal-spatial data, if released, can benefit the community in general.

Weakness:

One of the main contributions seem to be using a pretrained FPN for proposing object bounding boxes, which is claimed to be more generalizable. Are there comparisons of this intermediate representation to other vision pipelines such as mask predictions or object detections? I can imagine the recall rates for these methods for finding interesting objects would be lower.

The applicability of pretrained object detectors can be limited in some cases such as deformable or granular object manipulations. The region proposals seem to make the network focus less on the irrelevant parts, but the generality and robustness of the approach still need more experiments to verify. A depth camera might also be able to help filter out backgrounds in some cases.


**Summary Of Recommendation:**

This paper presents a well-thought and clearly written approach to imitation learning through object-centric representation and transformer backbones. The method outperforms baseline methods in simulation and in the real world. There are still several weaknesses regarding the general applicability and performance of the methods.

---

> ### Author Response · Authors · 2022-08-24
> **Response (1/3)**
>
> We thank the Reviewer for the thoughtful feedback! Here is our response to address the reviewer’s comments. All the line numbers here refer to the revised pdf to avoid line number mismatch from the original paper, and all the revised parts in the pdf are highlighted in red.
>
> > “One of the main contributions seem to be using a pretrained FPN for proposing object bounding boxes, which is claimed to be more generalizable. Are there comparisons of this intermediate representation to other vision pipelines such as mask predictions or object detections? I can imagine the recall rates for these methods for finding interesting objects would be lower.”
>
> The pre-trained RPN (pre-trained FPN) [1] we used is general for localizing objects from images, as we mentioned in Ln 156 - Ln 159. Such a generalization is supported by the impressive object detection results in the referenced work [1], where the authors showed that their RPN model could localize objects reliably in open-vocabulary datasets. RPN is also commonly used as an intermediate representation for other vision pipelines, including mask predictions [2] and object detections [3, 4] mentioned by the reviewer.
>
> For object detection, the downstream architectures following RPN are optimized for visual recognition tasks and thus have a large domain gap from our manipulation tasks which requires actionable features for continuous control. But RPN itself is optimized for localizing objects. Thus it is generally applicable to our manipulation task, as mentioned in Ln 166. As for mask predictions, it is more difficult to represent the positional information of masks to VIOLA as we did for object proposals. We believe it is worth investigating in the future.
>
> > “The applicability of pretrained object detectors can be limited in some cases such as deformable or granular object manipulations.”
>
> In this work, we focused on manipulating rigid and articulated objects but not deformable ones with more fine-grained structures. Deformable object manipulation is actively studied by a large body of works, such as [5, 6, 7, 8, 9]. Note that RPN is trained on LVIS and ImageNet datasets that contain many deformable object categories. Therefore, the pre-trained RPN can, in principle, crop out the deformable objects using proposal bounding boxes. Some experiments in VIOLA show the promise of extending to deformable or granular objects in VIOLA. For example,  VIOLA can successfully manipulate the coffee machine that is composed of articulated parts. Therefore, VIOLA has the potential to handle deformable objects as well, given their composite structures and RPN is capable to crop them out. We leave this for future work.
>
> > “The region proposals seem to make the network focus less on the irrelevant parts, but the generality and robustness of the approach still need more experiments to verify.”
>
> The region proposals prune out most of the non-object features based on the object priors from pre-trained RPN, which can be seen in Figure 5 visualization of object proposals on the **Make-Coffee** task.
>
> The generality and robustness of the approach can be supported by our experiments in six different tasks ranging from simulation to the real world, allowing VIOLA to outperform state-of-the-art baseline BC-RNN by 45.8% in average success rates. Such a result is based on performance not only in the canonical setup, but also in testing variations ranging from placement variations, distractors in the scenes to camera view jittering. The outperforming success rates of VIOLA in such a wide range of tasks and objects show the generality and robustness of our approach.
>
>
>
> > “ A depth camera might also be able to help filter out backgrounds in some cases.”
>
> This is a very good point! We agree with the reviewer that depth information could help filter out backgrounds. We want to clarify that we focus on the RGB images only in our setup, and we have mentioned this as a limitation of this work (Ln 330 - Ln 331). We only used RGB images in this work as the used RPN is pre-trained on ImageNet and LVIS detection datasets which are both datasets of RGB images. We expect to extend VIOLA to incorporate depth information in our future work.

---

> > ### Author Response · Authors · 2022-08-24
> > **Response (2/3)**
> >
> > > “The architecture design seems to contribute a lot to the success rates in the network ablation study. I wonder if the performance gap to BC-RNN is still big if we vary the number of demonstrations or increase the BC network size.”
> >
> > This is a very good question! We acknowledge that the architecture design contributes to the success rate, as shown by the comparisons of BC-RNN and the ablation model (+Temporal Positional Encoding) in Table 2. However, we also want to point out that the last three ablation models in Table 2 show our design of region proposal features improves the performance significantly besides the transformer architecture.
> >
> > We address the reviewer’s question on the number of demonstrations and network size. We ran additional experiments and found that: neither increasing the number of demonstrations nor increasing the neural network size would guarantee a smaller performance gap between VIOLA and BC-RNN. We present a detailed analysis in the following:
> >
> > *Vary the number of demonstrations*: We agree that if the number of demonstrations increases, it could improve the performance of BC-RNN, which is shown in [4] (see its Sec 4.6). But increasing the number of demonstrations does not necessarily lead to a smaller performance gap between BC-RNN and VIOLA. We ran additional experiments of training VIOLA and BC-RNN with ¼ and ½  of demonstrations (which are 25 and 50 demonstration trajectories, respectively). The performance gap between the two models becomes larger when increasing data from ¼ to ½ of demonstrations but becomes smaller again when increasing data from ½ to full demonstrations. This suggests that while more demonstrations increase the performance of both models, it does not necessarily lead to smaller performance gaps.
> >
> > We also see from the table that VIOLA constantly outperforms BC-RNN using different numbers of demonstrations. It suggests the effectiveness of VIOLA over BC-RNN. VIOLA outperforms BC-RNN because it uses the object proposal that allows the policy to reason over task-relevant image regions more effectively. Moreover, from the additional experiment, we can see that VIOLA has a leap in performance when using ½ of the demonstration datasets. This also shows that the VIOLA policy is significantly more effective than the BC-RNN model in learning robust policies from a small dataset.
> >
> >
> > |   Models  | ¼ of demonstration dataset | ½ of demonstration dataset | full dataset|
> > | -------------| ------------------------- | -------------------------|----------------------------|
> > | VIOLA     |   29 $\pm$ 4.0       |   73.5 $\pm$ 0.5        |   87.6 $\pm$ 1.1    |
> > | BC-RNN  |   14.0 $\pm$ 0.0    |    36.8 $\pm$ 0.0       |    62.8 $\pm$ 0.9   |
> >
> > *Increase the neural network size*: We ran additional experiments to show that increasing the neural network size of BC-RNN does not necessarily improve the performance. The following table compares the performance of BC-RNN using larger network sizes (by increasing the visual backbone size). We see that models using larger networks (both ResNet-34 and ResNet-50) both have 7% lower performance than the one using ResNet-18, thus even lower than the performance of VIOLA. We hypothesize that in imitation learning, larger visual backbones encode the features that are less efficient and generalizable than using smaller backbones for manipulation. In conclusion, increasing neural network size does not close the performance gap between BC-RNN and VIOLA.
> > |   Models  | ResNet-18              | ResNet-34             |  ResNet-50              |
> > | -------------| ------------------------- | -------------------------|----------------------------|
> > | BC-RNN  |  62.8 $\pm$ 0.9      |   55.7 $\pm$ 0.2    |    55.3 $\pm$ 6.9    |

---

> > > ### Author Response · Authors · 2022-08-24
> > > **Response (3/3)**
> > >
> > >
> > > > “Similarly, I wonder if the context information from spatial feature map, eye-in-hand image, and proprioception also provided to the simple BC baselines. I.e. how much the region proposals help with the transformer policy, v.s. How much the transformer policy outperforms vanilla networks.”
> > >
> > > All our baselines also use the workspace images, eye-in-hand images, and proprioception information, the same as VIOLA. We have revised the text to include such model details in Appendix A (See Ln 507 - Ln 511 in Appendix).  Therefore, our comparison experiments are fair.
> > >
> > > The contribution of transformer architecture can be seen from the comparison between BC-RNN and the ablation model “+Temporal Positional Encoding” in Table 2, which is equivalent to the transformer architecture.  However, the comparison of the vanilla transformer policy and the policy with full region features “+Region Positional Features” implies the further contribution of region proposals to task success rates other than the transformer architecture, giving a 6% increase in success rates.
> > >
> > > We thank the reviewer for asking this question so that we can enhance the clarity of the paper further.
> > >
> > >
> > > > “The fact that random erasing helps a lot in table 2 is unclear to me. Is it because that the policy is overfitting to the K patches and some random masking would help with generalization?”
> > >
> > > As mentioned in Ln 221, we used the data augmentation of random erasing in VIOLA to prevent policies from overfitting to specific region features. A comparison between ablation models (+Region Positional Features) and (+Random Erasing) in Table 2 shows that the random erasing helps the generalization. On the other hand, to show that the generalization helps the model not overfit to region features rather than to the whole image pixels, we did a preliminary study of using random erasing for BC-RNN, which takes the whole image pixels as input. The experiment shows that using random erasing lowers the performance of BC-RNN by about 10% from the one without random erasing, implying that random erasing does not help prevent models from overfitting to the whole image features. This result supports our unique data augmentation of random erasing for VIOLA.
> > >
> > > References:
> > >
> > > [1]Detecting Twenty-thousand Classes using Image-level Supervision. Zhou et al. 2021
> > >
> > > [2]Mask R-CNN. He et al. 2018
> > >
> > > [3]Faster R-CNN: Towards Real-Time Object Detection with Region Proposal Networks. Ren et al. 2016
> > >
> > > [4] What Matters in Learning from Offline Human Demonstrations for Robot Manipulation. Mandlekar et al. 2021
> > >
> > > [5] Learning Visible Connectivity Dynamics for Cloth Smoothing. Lin et al. 2021
> > >
> > > [6] Learning Closed-Loop Dough Manipulation Using a Differentiable Reset Module, Qi et al. 2022
> > >
> > > [7] FlingBot: The Unreasonable Effectiveness of Dynamic Manipulation for Cloth Unfolding. Ha et al. 2021
> > >
> > > [8] Dextairity: Deformable manipulation can be a breeze. Xu et al. 2022
> > >
> > > [9] Autonomously Untangling Long Cables. Viswanath et al. 2022

---

> ### Author Response · Authors · 2022-08-27
> **Follow-up Response to Reviewer 77ch**
>
> Dear Reviewer 77ch,
>
> We would like to thank the reviewer again for their efforts and time in providing thoughtful feedback and comments. We’ve revised the paper according to the reviewer’s suggestions and replied to all the questions and concerns. If the reviewer has further concerns regarding our work, we welcome any follow-up discussions. If the reviewer has no further concerns,  we sincerely ask the reviewer to consider raising the score. Thanks a lot!

---

### Official Review · Reviewer_9cZF · 2022-07-25

**Originality:** Good
**Technical Quality:** Good
**Clarity Of Presentation:** Good
**Impact:** 3

**Recommendation:**

Weak Accept: I recommend accepting the paper, but will not argue for my recommendation if the majority of other reviewers have a different opinion.

**Summary:**

The paper presents VIOLA, an object-centric imitation learning method for manipulation. The idea is that object- and task-specific knowledge should be attended to for a manipulation policy. To achieve this, VIOLA uses a pre-trained object detector to find regions of interest, which are fed into a transformer-based policy. The paper introduces three simulated tasks (sorting, stacking, kitchen) and three real-world tasks (plate+fork setting, bowl retrieval+setting, and making coffee). VIOLA is evaluated on these tasks along with 3 baseline imitation learning methods (BC, OREO, BC-RNN).

**Issues:**

I'm surprised at how small the object proposal patches are (visualized in figure 5). For example, I would consider the coffee maker as a whole to be a single object, but one patch cannot capture this. Would the method struggle to be scale-invariant (e.g. if the size of objects or distractors becomes much larger or smaller in the image)?

It appears as though Dining-PlateFork and Dining-Bowl were initially going to be a single sequential task. Was this the case? Was there a failure here due to its long-horizon nature?

L117-118: "Our goal is to learn the policy... that maximizes the expected return." This phrasing makes it sound like reinforcement learning methods will be evaluated, but to my knowledge, the reward is not being maximized anywhere (and there is no reward function defined for any task). Furthermore, I can't find a description of how successes are determined (in computing the success rate).

Some of the descriptions could be clearer (no need to respond to these):
* I don't understand what the global context feature is doing (L161).
* Figure 2 is overly complex, perhaps partially due to the complexity of the method and partially due to too many details being included in the figure.
* Is "sinusoidal position encoding of the temporal positions" (L174) common? I don't entirely follow the purpose of this here. Do we expect a sinusoidal nature in these manipulation tasks?

**Quality Of The Limitations Section:**

Additional details required

**Reviewer Expertise:**

3: The reviewer is fairly confident that the evaluation is correct

**Robotics Focus:**

Sufficient demonstration on hardware

**Strengths And Weaknesses:**

Strengths
* Object-centric information seems useful and a transformer gives a path for how to use such information.
* The evaluation strategy is thorough. I appreciate the out-of-distribution tests (placement, distractor, camera jitter) and the A/B testing paradigm to mitigate human bias (L283).
* The make coffee task seems challenging and linked to a realistic application.
* VIOLA-Patch (a variant of VIOLA without object-specific patches) is a great baseline to differentiate performance due to transformers.

Weaknesses
* The method is somewhat complex, which makes it challenging to determine the key aspect of VIOLA's success. Data augmentations are not all ablated or used consistently across baselines (unclear). I'm glad the random erasing was included in the ablation, but I'm not sure how much the other augmentations (color, pixel shifting, etc.) contribute to VIOLA's success. It would be nice if all baselines used the same augmentations, or if not, include motivation for why the augmentations are specific to the method. The eye-in-hand image should also be used consistently.
* Does K (the number of object proposals) need tuning for each environment? It would be more compelling to use the same K in simulation and the real world (just stick with either 15 or 20). L205-206 says "we set K=15 which are found to attain a high recall rate on real images" - but I'm not sure how this was selected. Can K be easily inferred from the demonstrations or was any online tuning done here?

**Summary Of Recommendation:**

The motivation for this work is strong, and I like the experimental setup. I have some concerns regarding the complexity of the method (see weaknesses above), but if these are addressed then I believe the work can be a strong contribution to CoRL.

---

> ### Author Response · Authors · 2022-08-24
> **Response (1/3)**
>
> We thank the reviewer for the thoughtful feedback and positive comments! Here is our response to address the reviewer's comments and questions. All the line numbers refer to the revised pdf (rather than the original paper), and all the revised parts in the pdf are highlighted in red.
>
> >  “Data augmentations are not all ablated or used consistently across baselines (unclear). I'm glad the random erasing was included in the ablation, but I'm not sure how much the other augmentations (color, pixel shifting, etc.) contribute to VIOLA's success.” and “ It would be nice if all baselines used the same augmentations, or if not, include motivation for why the augmentations are specific to the method.”
>
> Color jittering and pixel shifting are used consistently across all baselines. Both have shown benefits in performance and generalization in previous work for continuous control tasks [1, 2, 3]. Therefore, the two augmentations are in place to contribute to the performance of all the models.
>
> Random erasing, on the other hand, is our unique data augmentation design for VIOLA to prevent overfitting to specific region features, as mentioned in Ln 225 - Ln 226. For BC-like models such as BC-RNN, using random erasing hurts the model’s performance, yielding 52.0 $\pm$ 0.7 %. The success rate is lower than using BC-RNN without random erasing (62.8 $\pm$ 0.9). We suspect that random erasing hurts the BC-RNN performance because the data augmentation does not prevent models from overfitting to whole image pixels, which is the input of BC-RNN. With such a comparison result, we reported the higher performance of BC-RNN, which is trained without random erasing. This is also why we specifically made random erasing the main ablated factor for VIOLA to support our intuition that random erasing can prevent VIOLA from overfitting to specific region features.
>
> We’ve revised the text on the data augmentation details for VIOLA and baselines (see Ln 528 - Ln 532 in Appendix A).
>
>
> > “The eye-in-hand image should also be used consistently”
>
> We used the same sensor modalities across our approach VIOLA and all baselines: workspace RGB  images, eye-in-hand RGB images, and proprioceptive states. We have revised the text to include such model details for the baselines in Appendix (See Ln 507 - Ln 511 in Appendix A). We thank the reviewer for this question to help enhance the clarity of our paper!
>
> > “Does K (the number of object proposals) need tuning for each environment? It would be more compelling to use the same K in simulation and the real world (just stick with either 15 or 20). L205-206 says "we set K=15 which are found to attain a high recall rate on real images" - but I'm not sure how this was selected. Can K be easily inferred from the demonstrations or was any online tuning done here?”
>
> The choice of K=20 in simulation is based on its high recall rate, as mentioned in Appendix B.1. Its value is held constant across all the simulation tasks without further tuning.
>
> However, we did not choose K=20 in the real world because we found that RPN did not output 20 object proposals on some frames in the demonstrations. Even though the number of proposals did not reach 20, the RPN had cropped out nearly all the objects in the scene. Therefore, we ended up with K=15 for the real-world tasks. The value K was conveniently selected based on whether RPN reliably covers a sufficient number of object proposals from all the demonstration data.
>
> > “I'm surprised at how small the object proposal patches are (visualized in figure 5). For example, I would consider the coffee maker as a whole to be a single object, but one patch cannot capture this. Would the method struggle to be scale-invariant (e.g. if the size of objects or distractors becomes much larger or smaller in the image)?”
>
> Pre-trained RPN itself can output a proposal over the whole machine, but usually, its confidence score is not in the top K ranking, same as the case for the two examples in Figure 5. The RPN is scale-invariant to localize an object even if its size varies in the image. On the other hand, how much scale variation the policy can handle depends on the range of variations seen during demonstrations. This is because our image encoder is not designed to be scale-invariant in our imitation learning setup, and having huge changes in scale will result in a very different feature embedding. Large feature differences will result in severe distribution mismatch, resulting in failures of imitation learning policies. One potential solution to scale-invariance is to allow the policy to adapt online, as mentioned in our limitation section (see Ln 325 - Ln 327).

---

> > ### Author Response · Authors · 2022-08-24
> > **Response (2/3)**
> >
> > > “It appears as though Dining-PlateFork and Dining-Bowl were initially going to be a single sequential task. Was this the case? Was there a failure here due to its long-horizon nature?”
> >
> > We designed these two tasks not because we saw failures of the entire long-horizon task. Rather, we designed them to showcase the sequential execution of the two VIOLA policies that complete a sequentially composed task. With such a design, we show the promise of future extension to compose VIOLA policies learned from different subtasks to solve a complicated, sequential manipulation task.
> >
> > VIOLA can handle long-horizon tasks, as demonstrated in the **Make-Coffee** task. The average length of **Make-Coffee** demonstrations (700 timesteps) is nearly twice the average length of each dining task’s demonstration (424 and 400 timesteps, respectively). Our result shows that VIOLA can complete these tasks significantly better than the state-of-the-art baseline BC-RNN.
> >
> >
> > > “L117-118: "Our goal is to learn the policy... that maximizes the expected return." This phrasing makes it sound like reinforcement learning methods will be evaluated, but to my knowledge, the reward is not being maximized anywhere (and there is no reward function defined for any task). ”
> >
> > The MDP formulation describes the general problem of learning closed-loop visuomotor policies. We define the visuomotor policy learning problem in MDP following the convention of previous work on behavioral cloning for manipulation [4, 5, 6]. In such MDP, the reward function can be a sparse reward function conditioned on the goal specifications. Maximizing the expected return is considered to maximize the task success rates. Regardless of the learning algorithms used for policies, the MDP formulation of visuomotor policy learning remains the same. To make the paper more precise, we have revised the paragraph to emphasize that the MDP formulation describes the problem of learning a visuomotor manipulation policy in general (see Ln 126-128).
> >
> > The behavioral cloning algorithms, on the other hand, optimize policies by minimizing the distributional mismatch between policy outputs and demonstrated actions [10]. In practice, the minimization of distributional mismatch is considered a surrogate loss for visuomotor policy optimization when using behavioral cloning.  We’ve added a brief formulation of behavioral cloning after MDP formulation in (Ln 129 - Ln 132).
> >
> > > “Furthermore, I can't find a description of how successes are determined (in computing the success rate).”
> >
> > Thanks for pointing this out! We did miss out on the determination of success in the text. In the simulation, the success is determined by whether the object states in the simulator meet the pre-programmed goal function. For example, in the **Sorting** task, the goal function is programmed to check if the two boxes are both in the bin, and the simulation environment decides a rollout is successful only if the goal function returns True. In the real world, the success of a rollout is determined by whether the objects are in the goal configuration, which is implicitly specified in the given demonstrations. We have added this detail in Appendix C (see Ln 608 - Ln 615) to clarify this point.

---

> > > ### Author Response · Authors · 2022-08-24
> > > **Response (3/3)**
> > >
> > > > “I don't understand what the global context feature is doing (L161).”
> > >
> > > The global context feature here is to provide additional information based on workspace image observation, and the feature is passed into the transformer to help decide which subset of region features are more relevant to the task. To provide more insight here: with only the local information of features, it might be a policy that might be ambiguous because two different stages might have similar local features of objects. For example, in the **BUDS-Kitchen** task, during the stages of putting the pot on the stove and taking the pot off the stove, most of the objects’ features are very similar in locations and visual features, and it is hard to differentiate the two stages. In this case, the global context feature helps to resolve ambiguity from the workspace observations.
> > >
> > > > “Figure 2 is overly complex, perhaps partially due to the complexity of the method and partially due to too many details being included in the figure.”
> > >
> > > Our intention is for Figure 2 to faithfully present the pipeline of VIOLA. It illustrates the spatial-temporal part and how each feature is encoded at each timestep. If the reviewer has any suggestions on simplifying the figure while still satisfying its intended purpose, we are more than willing to improve it accordingly!
> > >
> > > > “Is "sinusoidal position encoding of the temporal positions" (L174) common? I don't entirely follow the purpose of this here. Do we expect a sinusoidal nature in these manipulation tasks?”
> > >
> > > The sinusoidal positional encoding is for better training of the transformer architecture rather than reflecting the nature of manipulation tasks. The sinusoidal positional encoding of the temporal positions was initially used in the original transformer paper [7] and has been adopted by other works on dynamics learning [8] and visual-language robotic navigation [9]. The insight behind the sinusoidal positional encoding is to allow the transformer to consider the relative positional information among tokens from input sequences. We use the sinusoidal positional encoding to have the transformer reasoning over the temporal relations of observation frames.
> > >
> > >
> > > References:
> > >
> > > [1]  Look closer: Bridging egocentric and third-person views with transformers for robotic manipulation. Jangir et al. 2022
> > >
> > > [2] Image Augmentation Is All You Need: Regularizing Deep Reinforcement Learning from Pixels. Kostrikov et al. 2021
> > >
> > > [3] What Matters in Learning from Offline Human Demonstrations for Robot Manipulation. Mandlekar et al. 2021
> > >
> > > [4] GTI: Learning to Generalize Across Long-Horizon Tasks from Human Demonstrations. Xu et al. 2020
> > >
> > > [5] Human-in-the-Loop Imitation Learning using Remote Teleoperation. Mandlekar et al. 2020
> > >
> > > [6] Generalization Through Hand-Eye Coordination: An Action Space for Learning Spatially-Invariant Visuomotor Control. Wang et al. 2021
> > >
> > > [7] Attention Is All You Need. Vaswani et al. 2017
> > >
> > > [8] Learning to Encode Position for Transformer with Continuous Dynamical Model. Liu et al. 2020
> > >
> > > [9] Hierarchical Cross-Modal Agent for Robotics Vision-and-Language Navigation. Irshad et al. 2021
> > >
> > > [10] A Divergence Minimization Perspective on Imitation Learning Methods. Ghasemipour et al. 2019

---

### Official Review · Reviewer_Cdbg · 2022-08-01

**Originality:** Good
**Technical Quality:** Very Good
**Clarity Of Presentation:** Good
**Impact:** 3

**Recommendation:**

Weak Reject: I recommend rejecting the paper, but will not argue for my recommendation if the majority of other reviewers have a different opinion.

**Summary:**

This paper proposed a model architecture for behavior cloning. The key element of the architecture is to feed region proposals from an RPN into a transformer. Results in simulation and real suggest this model works well.

**Issues:**

See Weaknesses discussion.

**Quality Of The Limitations Section:**

Limitations are addressed clearly

**Reviewer Expertise:**

5: The reviewer is absolutely certain that the evaluation is correct and very familiar with the relevant literature

**Robotics Focus:**

Sufficient demonstration on hardware

**Strengths And Weaknesses:**

# Strengths

- Cool real-world videos.
- Nice looking architecture figure.
- Results in sim and real suggest the model works pretty well.
- Figure 5 attention rankings are interesting.

# Weaknesses

- An inexperienced researcher would read this introduction and think that this paper has come up with the idea of using object centric representations in visuomotor policy learning. But of course this is not the case. See for example Transporter Networks, CoRL 2020, which isn’t an object centric method but reviewed different prior works in this area.
- This also goes for the title, “VIOLA: Object-Centric Imitation Learning for Vision-Based Robot Manipulation” which seems to suggest the authors have invented this idea. A more appropriate title might be “Region Proposal Transformers for Behavior Cloning” or something like that.
- The paper refers to its method as an algorithm but I think it should probably be referred to as a model architecture. The algorithm it’s evaluated with is behavior cloning.
- Why does section 3.1 talk about MDPs? There is no concept of rewards, or transition functions, or discount factors that is used in this paper. This paper just studies behavior cloning. The MDP paragraph copy paste is irrelevant to this paper. “Our goal is to maximize the expected return…” is not true, instead the method’s goal is to fit human demos.
- The details of the architecture are probably not sufficiently described in order to implement it from reading the paper.
- Why is [22] cited for doing behavior cloning with an RNN? This has been done well before [22], for example:
  - R. Rahmatizadeh, P. Abolghasemi, L. Bo ̈lo ̈ni, and S. Levine, “Vision-based multi-task manipulation for inexpensive robots using end-to-end learning from demonstration,” ICRA 2018
  - P. Florence, L. Manuelli, R. Tedrake, “Self-supervised Correspondence in Visuomotor Policy Learning,” RA-L 2019
- IL methods suffering from covariate shift has been studied at least since DAgger, Ross et al, 2011. Not a thing only from 2019 onwards, as it is cited here.
- Even specifically RNN for BC with a GMM (also called MDN) was done in Rahmatizadeh et al noted above.
- Overall I believe the academic question of the text should be more specifically “do region proposals help do more effective behavior cloning?” With this more precise framing of the text, I’m not sure the experimentation is very thorough for this question. The only comparison that gives a hint here is the Patch rather than RPN question.
- Although the real world demos are in some ways impressive looking, they don’t seem to require much closed loop feedback.


**Summary Of Recommendation:**

I am conflicted on this paper. On the one hand the robot demos are pretty cool, and it seems to work pretty well. Meanwhile, I listed several key weaknesses with the presentation and novelty of the paper in the Weaknesses section. I’m pretty borderline on this paper, I would currently lean reject but could consider moving up to accept if the authors address the weaknesses I listed.

---

> ### Author Response · Authors · 2022-08-24
> **Response (1/3)**
>
> We appreciate Reviewer Cdbg for the thoughtful and comprehensive feedback. Here is our response to address the reviewer's comments and questions. All the line numbers refer to the revised pdf (rather than the original paper), and all the revised parts in the pdf are highlighted in red.
>
> > "An inexperienced researcher would read this introduction and think that this paper has come up with the idea of using object centric representations in visuomotor policy learning. But of course this is not the case. See for example Transporter Networks, CoRL 2020, which isn't an object centric method but reviewed different prior works in this area."
>
> Thank you for this feedback. Our work is motivated by the challenge of object-centric learning in robotics, where the definitions of objects are often fluid and task-dependent for manipulation tasks. To tackle this challenge, we consider objects as disentangled visual concepts that inform the robot’s decision-making. We use object proposals from RPN trained on large-scale image datasets that can localize a diverse range of objects. Object proposals as priors have been used in previous work for tasks other than manipulation, such as object detection and segmentation [1, 2, 3], dynamics learning [4], and visual-language understanding [5, 6]. Inspired by these successes, we investigated through our work whether the general object proposals can be used as priors to help effective and robust imitation learning for vision-based manipulation.
>
> To better position our work in the literature, we reviewed in Section 2 the object-centric representations applied to various domains that share the same vein as ours: representing scenes into objects as disentangled visual concepts. We reviewed works that used object poses [7-9] or detectors [10–12] (bounding boxes) as object-centric representations for manipulation see Ln 101 - Ln 106). To provide a comprehensive review of representations for manipulation, we also reviewed other visual representations in manipulation, such as spatial attention [13-15] or affordance [16,17] (see Ln 98), and we consider keypoints and dense descriptors as part of the affordance based representations which were mentioned in the Transporter Networks paper.
>
> Overall, we did not intend to claim that we are the first to use object-centric learning for manipulation, but rather that we introduce a new model that incorporates object proposal priors as object-centric representation for imitation learning in vision-based manipulation. We revised the introduction (see Ln 40 - Ln 43, Ln 47 - 49) to make our claims more specific and added references to keypoints and dense descriptors in the category of affordance representations (see Ln98).
>
> > "This also goes for the title, "VIOLA: Object-Centric Imitation Learning for Vision-Based Robot Manipulation" which seems to suggest the authors have invented this idea. A more appropriate title might be "Region Proposal Transformers for Behavior Cloning" or something like that."
>
> As we explained above, this work investigates if we can use object proposals as object priors in visuomotor manipulation, specifically in imitation learning. We appreciate the reviewer’s suggestion to make the title more specific to our proposed approach. We’ve thus revised the title to “VIOLA: Imitation Learning for Vision-Based Manipulation with Object Proposal Priors.”
>
> > "The paper refers to its method as an algorithm but I think it should probably be referred to as a model architecture. The algorithm it's evaluated with is behavior cloning."
>
> We concede that referring to VIOLA as a model would help the paper's clarity. We initially refer to VIOLA as an algorithm as it has two components in its architecture: object-centric representations based on general object proposals and transformer-based policies. Even though we focus on behavioral cloning in our work, the policy learning algorithm can be extended to any other imitation learning and reinforcement learning algorithms.
>
> With the comment from the reviewer, we think referring to VIOLA as a model would be more precise. Therefore we replaced the behavioral cloning algorithm. We revised this in  Ln 51.

---

> > ### Author Response · Authors · 2022-08-24
> > **Response (2/3)**
> >
> > > "Why does section 3.1 talk about MDPs? There is no concept of rewards, or transition functions, or discount factors that is used in this paper. This paper just studies behavior cloning. The MDP paragraph copy paste is irrelevant to this paper. "Our goal is to maximize the expected return…" is not true, instead the method's goal is to fit human demos."
> >
> > The MDP formulation describes the general problem of learning closed-loop visuomotor policies. We define the visuomotor policy learning problem with MDPs following the convention of previous work on imitation learning for manipulation [18-20]. In such MDP, the reward function can be a sparse reward function conditioned on the goal specifications. Maximizing the expected return is considered to optimize the task success rates. Regardless of the learning algorithms used for policies, the MDP formulation of visuomotor policy learning remains the same. To make the paper more precise, we have revised the paragraph to emphasize that the MDP formulation describes the problem of learning a visuomotor manipulation policy in general (see Ln 126-128).
> >
> > The behavioral cloning algorithms, on the other hand, optimize policies by minimizing the distributional mismatch between policy outputs and demonstrated actions [21], which is exactly “fit human demonstrations” mentioned by the reviewer. In practice, the minimization of distributional mismatch is considered a surrogate loss for visuomotor policy optimization when using behavioral cloning. We’ve added a brief formulation of behavioral cloning after MDP formulation in (Ln 129 - Ln 132).
> >
> >
> > > "The details of the architecture are probably not sufficiently described in order to implement it from reading the paper."
> >
> > We included the implementation details of our models in both Ln 218 - Ln 228, and Appendix A. For further reproducibility, we have included the code in the supplementary zip file in the original submission. We promise to release it open-sourced later. If the reviewer still finds any specific missing implementation details, we would appreciate it and will add them accordingly.
> >
> >
> > > "Why is [22] cited for doing behavior cloning with an RNN? This has been done well before [22]" and "Even specifically RNN for BC with a GMM (also called MDN) was done in Rahmatizadeh et al noted above."
> >
> > We thank the reviewer for pointing out the missing citation. We cited Mandlekar et al. [22] (also [22] in the original pdf) initially because we followed their open-sourced implementation for comparison experiments. We agree with the reviewer that we should cite more works that also have done BC-RNN before [22]. We further did an extensive literature review on using BC-RNN before [22], and we’ve added additional references to the revision, including the two mentioned by the reviewer: [18, 23-25] (see Ln 262).
> >
> > > "IL methods suffering from covariate shift has been studied at least since DAgger, Ross et al, 2011. Not a thing only from 2019 onwards, as it is cited here."
> >
> > As VIOLA focused on deep imitation learning policies for vision-based manipulation, we initially cited the papers that discuss and address covariate shifts in deep imitation learning policies, especially in robotics. We agree with the reviewer that we could include more previous works on covariate shifts that are not limited to deep imitation learning policies. Therefore, we’ve added references that 1) initially discussed covariate shifts problems in imitation learning [26] and 2) approaches that first try to mitigate covariate shifts from the online setting (Dagger [27], which is mentioned by the reviewer, SafeDagger [28]) and the offline setting (ValueDice [29]) (see Ln 25).
> >
> > > "Although the real world demos are in some ways impressive looking, they don't seem to require much closed loop feedback."
> >
> > The tasks do require closed-loop feedback to complete reliably. For example, in the **Make-Coffee** task, closing the coffee machine could fail due to high friction of shutting down the lid or slipping of gripper fingers. In this case, the policy needs to observe the state and reactively generate actions to retry the closing. Such a closed-loop behavior can be found in videos such as [example 1](​​https://youtu.be/4FdFy9CJ8Is?t=127), [example 2](https://youtu.be/m2svU8J1TWM?t=21). Such a closed-loop requirement also exists in the **Dining-Bowl** task. For example, closed-loop feedback is required for the policy to either continue to rotate the table or pick up the bowl.

---

> > > ### Author Response · Authors · 2022-08-24
> > > **Response (3/3)**
> > >
> > > > "Overall I believe the academic question of the text should be more specifically "do region proposals help do more effective behavior cloning?" With this more precise framing of the text, I'm not sure the experimentation is very thorough for this question. The only comparison that gives a hint here is the Patch rather than RPN question."
> > >
> > > Our work is motivated by whether representing objects in the scenes into disentangled visual concepts could facilitate decision-making in vision-based manipulation (see Ln 40). RPNs have been pre-trained on huge image datasets to capture general priors of “objectness” across appearance variations and categories (which is mentioned in Ln 45). They are used in many downstream tasks, such as object detections and segmentation [1, 2, 3], dynamics learning [4], and visual-language understanding [5, 6]. Inspired by these prior works, we are interested in whether such a prior in the form of object proposals could improve the generalization of visuomotor policies in manipulation.
> > >
> > > Therefore, we would like to re-state our research question: **”How will object proposals serve as a prior for imitation learning in vision-based robotic manipulation?”**
> > >
> > > Our experiment results help answer this question by including an extensive comparison among our approach, VIOLA, and other baseline models: baselines without any object priors (BC, BC-RNN), baselines with unsupervised object priors (OREO), and object-agnostic region-based baseline (Patch-based). The comparison results suggest the use of object proposals improves the performance and robustness of behavioral cloning policies for vision-based manipulation.
> > >
> > >
> > >
> > > References:
> > >
> > > [1] Cascade r-cnn: Delving into high quality object detection. Cai et al. 2018
> > >
> > > [2] Objects as points. Zhou et al. 2019
> > >
> > > [3] Detecting twenty-thousand classes using image-level supervision. Zhou et al. 2022
> > >
> > > [4] Learning long-term visual dynamics with region proposal interaction networks. Qi et al. 2020
> > >
> > > [5] Uniter: Learning universal image-text representations. Chen et al. 2019
> > >
> > > [6] Vl-bert: Pre-training of generic visual-linguistic representations. Su et al. 2019
> > >
> > > [7] Deep object pose estimation for semantic robotic grasping of household objects. Tremblay et al. 2018
> > >
> > > [8] 6-dof pose estimation of household objects for robotic manipulation: An accessible dataset and benchmark. Tyree et al. 2022
> > >
> > > [9] Object-centric task and motion planning in dynamic environments. Migimatsu et al. 2020
> > >
> > > [10] Deep object-centric policies for autonomous driving. Wang et al. 2019
> > >
> > > [11] Graph-structured visual imitation. Sieb et al. 2020
> > >
> > > [12] Deep object-centric representations for generalizable robot learning. Devin et al. 2018
> > >
> > > [13] Transporter networks: Rearranging the visual world for robotic manipulation. Zeng et al. 2020
> > >
> > > [14] Learning to rearrange deformable cables, fabrics, and bags with goal-conditioned transporter networks. Florence et al. 2021
> > >
> > > [15] What and where pathways for robotic manipulation. Shridhar et al. 2022
> > >
> > > [16] Volumetric grasping network: Real-time 6 dof grasp detection in clutter. Breyer et al. 2021
> > >
> > > [17] Synergies between affordance and geometry: 6-dof grasp detection via implicit representations. Jiang et al. 2021
> > >
> > > [18] GTI: Learning to Generalize Across Long-Horizon Tasks from Human Demonstrations. Xu et al. 2020
> > >
> > > [19] Human-in-the-Loop Imitation Learning using Remote Teleoperation. Mandlekar et al. 2020
> > >
> > > [20] Generalization Through Hand-Eye Coordination: An Action Space for Learning Spatially-Invariant Visuomotor Control. Wang et al. 2021
> > >
> > > [21] A Divergence Minimization Perspective on Imitation Learning Methods. Ghasemipour et al. 2019
> > >
> > > [22] What Matters in Learning from Offline Human Demonstrations for Robot Manipulation. Mandlekar et al. 2021
> > >
> > > [23] Vision-based multi-task manipulation for inexpensive robots using end-to-end learning from demonstration. Rahmatizadeh et al. 2018
> > >
> > > [24] Self-supervised Correspondence in Visuomotor Policy Learning. Florence et al. 2019
> > >
> > > [25] Learning to Play by Imitating Humans. Dinyari et al. 2020
> > >
> > > [26] A Reduction of Imitation Learning and Structured Prediction to No-Regret Online Learning. Ross et al. 2011
> > >
> > > [27] Query-Efficient Imitation Learning for End-to-End Autonomous Driving. Zhang et al. 2016
> > >
> > > [28]Imitation Learning Via Off-Policy Distribution Matching. Kostrikov et al. 2019
> > >
> > > [29] Efficient Reductions for Imitation Learning. Ross et al. 2010

---

> ### Author Response · Authors · 2022-08-27
> **Follow-up Response to Reviewer Cdbg**
>
> Dear Reviewer Cdbg,
>
> We would like to thank the reviewer again for their efforts and time in providing thoughtful feedback and comments. We’ve revised the paper according to the reviewer’s suggestions and replied to all the questions and concerns. If the reviewer has further concerns regarding our work, we welcome any follow-up discussions. If the reviewer has no further concerns,  we sincerely ask the reviewer to consider raising the score. Thanks a lot!

---

### Author Response · Authors · 2022-08-27
**Thanks to all reviewers and meta-reviewers!**

Dear reviewers and meta-reviewers,

We are very grateful for your time and efforts in providing us with great and constructive feedback to strengthen our paper even further. We appreciate that the reviewers have found our real robot experiments interesting and impressive, our approach is effective for robust generalization, and the ablation studies are useful.

In this rebuttal, we have revised the paper and conducted additional ablation studies to address some of the reviewers’ concerns. We respond to each reviewer’s questions and concerns individually below, and we’ve attached the PDF of the revised paper, where all the changes are highlighted in red. We welcome any follow-up discussions if you have any further concerns.

---

### Meta-Review · Area_Chair_Nc1w · 2022-08-15

**Recommendation:** Accept (Poster)
**Confidence:** 3

**Metareview:**

Below is a summary of the strengths and weaknesses of the paper, according to the reviewers.

Strengths:
- The use of transformers is interesting for attending to task-relevant parts of the image.
- Results demonstrate robustness across viewpoints, initialisations, and distractors (although there is concern that the distractors are not that significant).
- The ablation studies are very useful.
- The real-world demos are interesting and impressive.

Weaknesses:
- The paper appears to makes overly grand claims about introducing object-centric representations for imitation learning.
- The baselines do not use the same data augmentations as the proposed method, making it an unfair comparison.
- The use of pre-trained object detectors could make the method impractical for granular or deformable objects.

In the rebuttal, please address the above weaknesses, as well as the other concerns and questions raised by the reviewers.

----------

Update after rebuttal:

Following the rebuttal, opinions were still divided on whether the paper should be accepted. There is a consensus that the overall idea is sensible and performs well, with strong real-world demos. But there is still concern about what we can really learn from this paper, other than the strength of transformers and region proposals. However, some of the questionable claims made in the original paper have now been rephrased in an updated paper. After a private discussion, the opinion amongst the reviewers is generally more positive than negative, and I believe that the simplicity of the method, together with the compelling real-world demo, would be appealing to the community.

**Best Paper Nomination:**

No

---

> ### Author Response · Authors · 2022-08-24
> **Response to the Meta Reviewer**
>
> We would like to thank all the reviewers and the meta-reviewer for their high-quality reviews of our paper. We really appreciate all the detailed and insightful comments. We revised the paper and the appendix (all changes highlighted in red) according to the reviewers’ comments. We have also replied to each reviewer’s comments and concerns. We reply to the meta-reviewer’s comments as follows:
>
> > “The paper appears to makes overly grand claims about introducing object-centric representations for imitation learning.”
>
> Thank you for pointing this out. We’d like to clarify that our contribution is not being the first to use object-centric representations but rather introducing a new model that incorporates object proposal priors as object-centric representation for imitation learning in vision-based manipulation.
>
> Our work is motivated by the challenge of object-centric learning in robotics, where the definitions of objects are often fluid and task-dependent for manipulation tasks. To tackle this challenge, we consider objects as disentangled visual concepts that inform the robot’s decision-making. We use object proposals from RPN trained on large-scale image datasets that can localize a diverse range of objects.
>
> We've revised the introduction and the related work to enhance our clarity and tone down the claims. More specifically, we included additional discussion of previous works on object-centric imitation learning in manipulation (Ln 41-43) to give readers more context about the background and how our work differs from previous works. Then we added Ln 47 - 49 to emphasize the specific idea of using object proposals. In the original version, we already did a comprehensive literature survey covering object-centric representations in manipulation, such as detection bounding boxes, 6D poses  (Ln 101 - 106), and other related representations, such as spatial attention and affordance (Ln 90-100). We further added discussions on other types of object-centric priors for manipulation that do not use general object proposals (Ln 83 - 85) and additional reference that is based on affordance-based predictions (keypoints and dense descriptors, see Ln 98).
>
> > “The baselines do not use the same data augmentations as the proposed method, making it an unfair comparison.”
>
>
> As explained in response to Reviewers 9cZF, 77ch, we clarify that we used the same color jittering and pixel shifting consistently across all the models in our experiments (see Ln 224 - Ln 225). As for random erasing, it is specifically designed for VIOLA to avoid overfitting to region features. We observe a 10% drop in task success rates for BC-RNN if random erasing is used (in our response to Reviewer 77ch). Therefore, we consider the comparison among models presented in our experiments fair.
>
> > “The use of pre-trained object detectors could make the method impractical for granular or deformable objects.”
>
>
> In this work, we focused on manipulating rigid and articulated objects but not deformable ones with more fine-grained structures. Deformable object manipulation is actively studied by a large body of works, such as [1, 2, 3, 4, 5]. Note that RPN is trained on LVIS and ImageNet datasets that contain many deformable object categories. Therefore, the pre-trained RPN can, in principle, crop out the deformable objects using proposal bounding boxes. Some experiments in VIOLA show the promise of extending to deformable or granular objects in VIOLA. For example,  VIOLA can successfully manipulate the coffee machine that is composed of articulated parts. Therefore, VIOLA has the potential to handle deformable objects as well, given their composite structures and RPN is capable to crop them out. We leave this for future work.
>
> References:
>
> [1] Learning Visible Connectivity Dynamics for Cloth Smoothing. Lin et al. 2021
>
> [2] Learning Closed-Loop Dough Manipulation Using a Differentiable Reset Module, Qi et al. 2022
>
> [3] FlingBot: The Unreasonable Effectiveness of Dynamic Manipulation for Cloth Unfolding. Ha et al. 2021
>
> [4] Dextairity: Deformable manipulation can be a breeze. Xu et al. 2022
>
> [5] Autonomously Untangling Long Cables. Viswanath et al. 2022